# Overweight and Fertility: What We Can Learn from an Intergenerational Mouse Obesity Model

**DOI:** 10.3390/ijerph19137918

**Published:** 2022-06-28

**Authors:** Dušan Fabian, Janka Kubandová-Babeľová, Martina Kšiňanová, Iveta Waczulíková, Kamila Fabianová, Juraj Koppel

**Affiliations:** 1Centre of Biosciences, Institute of Animal Physiology, Slovak Academy of Sciences, Šoltésovej 4/6, 040 01 Košice, Slovakia; kubandova@saske.sk (J.K.-B.); koppel@saske.sk (J.K.); 2Centre of Biosciences, Institute of Molecular Physiology and Genetics, Slovak Academy of Sciences, Dúbravská cesta 9, 840 05 Bratislava, Slovakia; martina.ksinanova@savba.sk; 3Faculty of Mathematics, Physics and Informatics, Comenius University, Mlynská dolina F1, 842 48 Bratislava, Slovakia; iveta.waczulikova@fmph.uniba.sk; 4Biomedical Research Center, Institute of Neurobiology, Slovak Academy of Sciences, Šoltésovej 4/6, 040 01 Košice, Slovakia; fabianova@saske.sk

**Keywords:** overweight, female mouse, ovulation yield, fertilization rate, preimplantation development

## Abstract

The aim of this study was to evaluate the effects of being overweight on the ability to conceive, fertilization rate, and in vivo development of embryos in regularly cycling, spontaneously ovulating, and naturally mated female mice. The study was based on statistical analysis of data collected during 14 experiments with identical design, performed on 319 control and 327 obese mice, developed in an intergenerational model of obesity induction which eliminates the impact of aging and high-fat feeding. Six-week-old mice with a vaginal sperm plug were slaughtered on embryonic days 2, 3, or 4, and the flushed contents of the oviducts and uteri were assessed by stereomicroscopy. The results showed no association between being overweight and the proportion of ovulating or fertilized females. On the other hand, a strong association was found between being overweight and ovulation yield. On embryonic day 2, significantly higher numbers of eggs were recovered from the oviducts of fertilized obese mice. Maternal overweight status was also associated with higher developmental capacities of preimplantation embryos. In conclusion, contrary to studies based on the high-fat-diet model, in female mice fed regular chow, being overweight was associated with an increased ovulation quota and higher developmental rate of fertilized oocytes. Being overweight did not impact ability to conceive. On the other hand, as documented in our previous studies, the quality of oocytes and blastocysts recovered from overweight mice developed in an intergenerational model of obesity was low.

## 1. Introduction

Assessment of the impact of being overweight on the ability to conceive and early development of conceptus in women has been carried out in numerous systematic studies. However, many of these have produced conflicting outcomes, documenting no or a negative effect of being overweight on particular reproductive parameters (reviewed in [1,2,3,4]). To minimize the impact of variable factors on the results, researchers have turned to animal models. Experimental studies on mice with diet-induced obesity have confirmed an association between female obesity and reproductive disorders in the peri-conception period [5,6,7,8,9,10]. However, some of these studies produced inconsistent results as well.

To respect specifics of reproductive research, an intergenerational model of diet-induced obesity was established for the production of outbred mice that are overweight [11]. Mice developed in this intergenerational model do not show the same massive fat accumulation as genetically modified obese mice or obese mice produced by means of the standard high-fat-diet model [12]; however, they are able to develop a somatic condition characterized by significantly higher body weight and body fat and increased plasma glucose, insulin, and leptin levels at the age of 35 days [13,14,15,16], i.e., at the time characterized by the highest ability to conceive in selected strain [17,18]. Furthermore, the acquired characteristics of body condition show long-lasting stability, i.e., there is no decrease in body weight which usually accompanies the removal of the high-fat diet. Additionally, since both control and overweight females are fed the same type of diet during mating and conceptus development (the only difference is in their caloric intake [13]), this model enables researchers to differentiate the impact of body condition and actual nutrition on reproductive parameters.

So far, the intergenerational model of diet-induced obesity has been used to evaluate the effect of female overweight status on various qualitative parameters of ovarian cells, oocytes, and early embryos, such us lipid deposition, metabolic activity, hormonal activity, apoptosis incidence, gene expression, and DNA methylation [11,14,15,16,19,20,21,22]. The results have helped to clarify some of the unknown aspects of this relationship. However, the impact of being overweight on quantitative reproductive parameters remains to be evaluated.

In general, quantitative reproductive parameters are characterized by high individual variability in females. Furthermore, they might be influenced by a number of external factors, such us season, health, stress, libido, or quality of sperm. Not all of these factors can be eliminated when planning experiments on living animals; however, their impact on results can be minimized through the use of a large sample size.

The aim of the current study was to evaluate the effects of female overweight status on the ability to conceive, ovulation quota, fertilization rate, and in vivo development of preimplantation embryos by means of statistical analysis of data collected during 14 experiments with identical design performed between the years 2011 and 2020 on control and obese mice using an intergenerational obesity model.

## 2. Materials and Methods

### 2.1. Experimental Design

#### 2.1.1. Production of Overweight Female Mice

The experiments were performed on mice (*Mus musculus*) of the outbred ICR (CD-1 IGS) strain (Velaz, Prague, Czech Republic). Overweight females were produced using a breeding protocol based on over-nutrition of animals during intrauterine and early postnatal development (Figure 1). Briefly, hormonally synchronized female mice of the parental generation were mated with males of the same strain and then randomly divided into two groups: Dams in the control group were fed a standard pellet diet (M1, Ricmanice, Czech Republic, 3.2 kcal/g) ad libitum, and dams in the experimental group were fed a standard pellet diet (M1) and Ensure Plus high-energy nutritional product (Abbott Laboratories, Lake Bluff, IL, USA, 1.5 kcal/mL), also ad libitum. To provide adequate nutrition to the delivered pups, the litter size was adjusted on the eighth day after birth to up to 10 pups per nest. After weaning, all female mice of the filial generation were fed the standard pellet diet only (M1). All animals were kept at 22 ± 1 °C on a 12 h light/dark cycle (6 am to 6 pm) with free access to food and water.

On day 34, females of the filial generation were individually scanned using EchoMRI (Whole Body Composition Analyser, Echo Medical System, Houston, TX, USA) and allocated into four groups: females delivered from control dams were classified as normal controls with physiological body weight (±20 g) and body fat (7–8%) or lean controls spontaneously displaying decreased body weight and body fat (<7%); females delivered from experimental dams were classified as obese mice with significantly elevated body weight (±24 g) and fat (>11%), or obesity-induction resistant experimental mice with physiological body weight and slightly elevated body fat (8–11%). In the following reproductive experiments, only control mice with physiological body weights and body fat and obese experimental mice with significantly elevated body weight and body fat were used.

#### 2.1.2. Mating of Female Mice

At day 35, spontaneously ovulating control and obese female mice were mated with males of the same strain (Velaz, 10 to 16 weeks old) for one or more nights. In each experiment, a cohort of 50 males with inbreeding-excluding origin and previously proven fertility was used for mating purposes. To minimize the impact of paternal obesity or stress, males were fed the standard pellet diet only and housed individually. Successful mating was confirmed by the identification of a vaginal sperm plug every morning.

#### 2.1.3. Recovery of Ova and Embryos

Mated dams from both groups were sacrificed using cervical dislocation and were subjected to ova/embryo isolation on embryonic day 2 of pregnancy (ED2, approx. 32 h after presumed ovulation [15,20]), on embryonic day 3 of pregnancy (ED3, approx. 56 h after presumed ovulation [16]), and on day 4 of pregnancy (ED4, approx. 80 h after presumed ovulation [11,15,16]). Recovery was performed by flushing oviducts and uteri using a flushing-holding medium [23] containing 1% bovine serum albumin (BSA, Sigma-Aldrich, St. Louis, MO, USA). Immediately after isolation, the collected material was subjected to stereomicroscopic classification (Nikon SMZ 745T, Nikon, Tokyo, Japan), and the numbers of isolated ova/embryos were determined. Data gathering was performed between the years 2011 and 2020. During this period, ova/embryo isolation was performed eight times on ED2, three times on ED3, and five times on ED4.

#### 2.1.4. Classification of Female Mice

Female mice with a vaginal sperm plug were classified as “mated females”. Female mice with developing corpora lutea and at least one ova or embryo in the reproductive tract were classified as “ovulated females”. Female mice with at least one ≥2 cell embryo in the reproductive tract were classified as “fertilized females”.

#### 2.1.5. Classification of Ova and Embryos

Successful fertilization was confirmed by the presence of ≥2 cell embryos in the maternal reproductive tract. Flushed ova with normal morphology were classified as unfertilized oocytes. Ova showing abnormal morphology (e.g., absent polar body, highly translucent cytoplasm with non-homogenous structure, cytoplasmic autolysis or fragmentation) were classified as unfertilized and degenerated oocytes. Embryos displaying 2 to 16 cells were classified as cleaving-stage embryos, embryos displaying >16 cells were classified as morulas, and embryos displaying a blastocoele cavity were classified as blastocysts. Embryos with abnormal morphology (showing cytoplasmic fragmentation, autolysis, or containing remnants of degraded biological material only) were classified as degenerates.

### 2.2. Statistical Analysis

Statistical analysis was performed using StatsDirect 3.2.8 software (StatsDirect Ltd., Cheshire, UK).

Categorical variables are presented as counts and relative frequencies. Data showing departures from normality are presented as medians with the interquartile range (IQR). Descriptive and univariate analyses were performed in order to identify important reproductive characteristics associated with maternal obesity.

Fisher’s exact test was conducted to test for an association between maternal overweight status and proportion of ovulating and fertilized females on day 2, 3, and 4 after mating. Variables are quoted with the corresponding odds ratio (OR) along with the respective 95% confidence interval (CI).

Poisson regression models were fitted for the outcomes of the number of recovered ova and embryos on embryonic day 2, 3, and 4; and for the outcomes of the number of recovered fertilized ova (≥2 cell embryos) and unfertilized ova (oocytes and degenerates) on ED 2, the number of quickly developing embryos (≥8 cell) and slowly developing embryos (<8 cell) and ova on ED 3, and the number of blastocysts and slowly developing embryos (morulas and <16 cell embryos). The outcomes were regressed against the presence of maternal overweight status in donor mouse females. The selected model variables are quoted with the corresponding regression coefficient (B), along with the respective 95% confidence intervals (CI).

The significance level was set at *p* < 0.05.

## 3. Results

This study was performed on 319 control and 327 obese female mice showing the presence of a vaginal sperm plug after spontaneous ovulation and overnight mating with the males (Figure 2).

The results of Fisher’s exact test showed no association between maternal overweight status and the **proportion of ovulating females** or between maternal overweight status and the **proportion of fertilized females** on any day of early pregnancy (Table 1). Independently of maternal body condition, ova or embryos were recovered from approximately 90% of the mated females (Table 1). On embryonic day 2, the proportion of fertilized females (i.e., females containing at least one cleaving embryo) in the group of obese mice tended to be lower than in the control group (odds ratio 0.57); however, the difference was not significant (*p* = 0.153).

The results of the Poisson regression model indicated a strong association between maternal overweight status and **total number of ova** recovered from fertilized mice (Table 2): On embryonic day 2, a significantly higher number of ovulated ova was isolated from obese females than from control females (95% CI 1.05–1.21; *p* = 0.0008). Further analysis of ovulated ova showed that maternal obesity was also associated with a significant increase in the number of fertilized ova (≥2 cell embryos; 95% CI 1.04–1.20; *p* = 0.0026) and a non-significant increase in the number of unfertilized ova (oocytes and degenerates; 95% CI 0.98–1.77; *p* = 0.062) (Table 3). On embryonic day 3, higher numbers of ova/embryos were flushed from the obese mice too; however, in this case, the difference between obese mice and the control group did not reach statistical significance (*p* = 0.13; Table 2). On embryonic day 4, no difference in the numbers of recovered embryos was recorded between groups (*p* = 0.17 for ED4; Table 2).

The results of the Poisson regression model indicated an association between maternal overweight status and **developmental capacities of embryos** recovered from fertilized mice (Table 4 and Table 5): Oviducts/uteri of obese female mice contained significantly higher numbers of embryos with ≥8 cells on ED3 (95% CI 1.16–1.76; *p* = 0.0007) and a significantly higher number of blastocysts on ED4 (95% CI 1.06–1.32; *p* = 0.0027). In counterpart, maternal obesity was associated with a non-significantly decreased number of slowly developing embryos on ED3 (<8 cell embryos, oocytes, and degenerates; *p* = 0.255) and a significantly decreased number of slowly developing embryos on ED4 (morulas, 16 cell embryos, and degenerates; *p* = 0.0014).

In fertilized female mice, the majority of oocytes recovered on ED2 and ED3 showed normal morphology. The occurrence of degenerated oocytes was very rare (control group: 0.013% on ED2, 0.008% on ED3; overweight group: 0.008% on ED2, 0.005% on ED3). On ED4, the proportion of degenerated oocytes/embryos reached 0.047% in the control group and 0.042% in the overweight group. No statistical difference between the groups was detected on any embryonic day.

## 4. Discussion

### 4.1. Ovulation Yield

Analysis of data collected from 14 independent experiments showed that being overweight, in spontaneously ovulating six-week-old female mice, might have a positive effect on their ovulation rate (Table 2).

Similarly, significantly increased numbers of ovulated oocytes have been previously documented in female mice fed a high-fat diet and displaying elevated body weight, higher amounts of abdominal fat, and insulin resistance at the age of 21 weeks [8]. On the other hand, in that study, a higher number of obese mice were classified as anovulatory (6 from 15, i.e., 40%). Although the proportion of mated female mice with empty oviducts varied between particular experiments in our study, the overall statistical analysis of our data revealed no difference in their frequency between obese and control mice (it did not exceed 9% in either group; Table 1). The contradiction in these findings might be explained by dissimilarities in animal numbers or in animal age, which would potentiate the negative effect of obesity on ovarian function. On the other hand, since the highest number of “empty” females was recorded after the first night of mating, we might hypothesize that the absence of embryonic material in the oviducts more likely resulted from copulation with females in the non-ovulation phase of the estrous cycle rather than from ovarian dysfunction. Naturally, the occasional occurrence of anovulation or fallopian tube obstruction cannot be excluded even in six-week-old mice; however, this ethology was not evaluated.

Our observations contradict the results of population-based studies in humans. In women, the association between obesity and anovulatory infertility has been documented several times (reviewed in [24]). However, much of this ovulatory dysfunction is likely confounded by a diagnosis of polycystic ovary syndrome. The distribution of body fat is also important, because not the intraabdominal fat but the subcutaneous fat has been shown to be associated with anovulation [25]. Nevertheless, to identify specific factors explaining contradictions between findings in mice and humans is very difficult, because of numerous differences in both reproductive physiology (length of estrous cycle, number of ovulated eggs, etc.) and the methodological approach (contrary to the experimental animals, in women, the heterogeneity of studied groups is usually very high: they differ in age, genetic background, dietary habits, healthcare, lifestyle, exercise, etc.). Furthermore, in both species, the continuous monitoring of ovulation activity is technically very demanding.

The majority of other previous experimental studies showed no effect of female obesity on ovulation rate: 7- to 20-week-old mice with diet-induced obesity showed similar numbers of recovered oocytes [10,26,27], zygotes [28], or 2 cell embryos, on average [7], as controls. However, in all of these studies, female mice were subjected to ovarian stimulation with gonadotropins before ova/embryo recovery, and such treatment would overlap with the effects of obesity. In contrast, a positive relationship between obesity and follicular activity has been reported in 34-week-old overfed gilts displaying increased body weight and back fat [29].

The mechanism of the positive effect of being overweight on the ovulation rate was not evaluated in our study. However, based on previous findings, we can hypothesize that positive energetic balance accompanied by increased endocrine activity of pancreas and adipose tissue probably influences gonadotropin secretion in the pituitary gland and subsequently stimulates the development and ovulation of a higher number of follicles in the ovaries of obese mice. It has been shown that increased circulating insulin levels are associated with increased luteinizing hormone (LH) secretion through elevated expression and secretion of gonadotropin-releasing hormone (GnRH) in mice [30,31]. Furthermore, studies on obese mice have shown that leptin increases hypothalamic insulin sensitivity [32] and potentiates the effect of insulin on GnRH secretion [30]. The requirement of endogenous leptin for activation of hypothalamic GnRH secretion and physiological GnRH pulse generation through kisspeptin or glutamate has been described as well [33].

### 4.2. Fertilization Rate

On the other hand, it has been proposed that due to LH hypersecretion and an increased LH:FSH ratio, the resumption of oocyte maturation in developing follicles is frequently impaired by obesity in women [1,3]. Oocytes retrieved from women with BMI >25 are usually of lower quality than those retrieved from women with a BMI of 20–25; i.e., the percentage of oocytes at germinal stages, postmature oocytes, and oocytes with fractured zona is significantly higher in the former [34]. Improvement in the pregnancy failure rate after the use of donor oocytes instead of autologous oocytes also indicates compromised oocyte quality in obese women [35].

Similar findings have been documented in animal experimental studies: mice with diet-induced obesity showed increased occurrence of abnormal morphology of mitochondria, elevated levels of reactive oxygen species, and epigenetic modifications in GV oocytes [7,12]; a lower percentage of mature germinal vesicle breakdown (GVBD) oocytes in follicles [12,36]; and a higher incidence of spindle and chromosome alignment defects in matured MI and MII oocytes [7,12]. Moreover, significantly higher rates of meiotic spindle and mitochondrial defects were found in oocytes of “diet reversal mice” exhibiting normalization of weight, glucose utilization, and cholesterol levels eight weeks after switching from a high-fat diet to regular chow [9]. Finally, matured oocytes recovered from obese mice developed in an intergenerational model showed significantly lowered deposits of neutral lipids in the cytoplasm [19], higher DNA methylation in the nuclear area [16], and significantly lower expression of Glutathione Peroxidase 8 [37].

Interestingly, the above stated facts did not affect the overall incidence of fertilization in overweight female mice in the current study: There was no difference in the average percentage of ≥2 cell embryos flushed from the oviducts of obese (93.36 ± 1.25%) and control dams (94.22 ± 1.23%) on embryonic day 2 (*p* > 0.05, chi-square test). These results are in accordance with previously published findings recorded on day 1 of pregnancy: At 11 h after supposed spontaneous ovulation, successful fertilization was confirmed in 70.34% and 79.84% of oocytes recovered from obese and control females, respectively [19]. On the other hand, obese mice developed in an intergenerational model produced a higher number of ovulated oocytes with abnormal morphology (i.e., with absent polar body, cytoplasmic autolysis, or fragmentation, etc.).

Our findings contradict the outcomes of other in vivo studies documenting decreased fertilization rate of oocytes originating from mice fed a high-fat diet [7,26] and rabbits fed with a hyperlipidic hypercholesterolemic diet [38]. This suggests that fertilization disorders are more likely to be connected to diet alterations than obesity phenotype [39] (i.e., consumption of a diet with elevated fat vs. standard diet during the peri-conception period). An obesity-independent negative effect of elevated dietary fat on fertilization rate has been documented previously [40]. Furthermore, in later studies, a negative effect of diet could be potentiated by hormonal treatment: It has been shown that ovarian stimulation with gonadotropins decreases the fertilization abilities of mouse oocytes as well [41].

In humans, in vitro fertilization rates have been inconsistently linked to maternal BMI too (reviewed in [42]).

### 4.3. Developmental Rate

The results of the current study showed that the cleavage rate for preimplantation embryos developing in the reproductive tract of obese dams was significantly higher than that in controls (Table 4 and Table 5). A shorter time post-insemination for the morula stage to be reached was also documented in embryos from women with a BMI > 25 [43] and over-conditioned cows with body condition score (BCS) > 4 [44].

Based on these observations, we might hypothesize that the oviducts of obese females probably provide a beneficial microenvironment for embryonic cell growth. Unfortunately, there are limited data on the composition of oviductal fluid in obese women or overweight animals to directly support this hypothesis [28,45,46]. However, the significantly elevated secretion of progesterone, a critical hormone for the development of mouse embryos [47], from in vitro cultured post-ovulatory ovaries of obese mice documented in our recent study [22] corresponds with this hypothesis.

On the other hand, the quality of such “quickly derived” embryos is disputable. As shown in our previous studies, blastocysts recovered from obese mice displayed significantly elevated incidence of apoptotic cell death [11], altered gene expression (higher BAX/BCL2L2 ratio, higher expression of insulin receptor [16], and higher expression of GLUT4, an insulin-responsive glucose transporter [15]), and development of insulin resistance [16]. Furthermore, the origin of embryos (from obese vs. control dams) significantly altered their overall metabolic activity in vitro, i.e., their metabolomic profile assessed by means of Raman spectroscopy [20], their response to leptin and insulin during development in vitro [15,16], and their sensitivity to oxidative stress during development in vitro [37]. Similarly, in human in vitro developed blastocysts, fewer cells in the trophectoderm and poor glucose uptake were demonstrated in a group of obese women [43]. Blastocysts recovered from over-conditioned cows also displayed a higher dead cell index, lower cell numbers, lower number of transcriptionally active nucleoli, and elevated incidence of mitochondrial vacuolization [44,48].

Lower quality of blastocysts may not necessarily lead to implantation failure; however, an altered genomic or metabolic profile would have a significant impact on the further development of the conceptus or growth of the offspring. The documented fetal growth retardation [7,27] and decrease in body weight and fat in newborn mice [19] delivered from obese dams might result from such changes. Epidemiology studies in humans also suggest that maternal obesity increases the tendency among offspring to develop obesity and insulin resistance later in life, as a result of “nutritional programming” during gestation [49,50,51].

Nevertheless, our finding of a positive effect of maternal overweight status on the cleavage rate of preimplantation embryos in vivo contradicts the outcome of two previous studies on 12- and 19-week-old mice showing no differences, or even a slight decrease, in developmental capacities of growing embryos recovered from dams with diet-induced obesity [27,28]. Once again, the discrepancy might be explained by the use of a high-fat diet, older animals, and gonadotropins in the experimental design of previous research. Since there are data documenting a significant delay in morula formation, blastocoele formation, or zona pellucida lysis in mice of various strains after ovarian stimulation [21,52], we suppose that gonadotropin treatment per se could obscure the beneficial impact of obesity.

Published data on the effect of being overweight on the developmental dynamics of recovered zygotes or 2 cell embryos in vitro are controversial as well: some authors have documented no effect [20,26], whereas others have documented reduced ability of ova originating in obese animals to reach higher stages of development [8]. In humans, similar contradictory findings have been reported [43,53]. Still, these findings indicate the disappearance of any supporting effect of the maternal environment on embryonic development after the transfer of zygotes/embryos to equivalent artificial conditions in vitro.

## 5. Conclusions

The results of this “population study” performed on pubertal mice developed in an intergenerational model of obesity showed no impact of female overweight status on the frequency of spontaneous ovulation or natural fertilization. On the other hand, female overweight status was strongly associated with elevated ovulation quota. Furthermore, data on the proportion of fertilized and unfertilized ova recovered from fertilized dams suggest that when negative effects of superovulation, aging, and high-fat consumption are eliminated, obesity-induced alterations in qualitative parameters of matured oocytes do not necessarily have to be linked to fertilization disorder. However, a higher ovulation yield did not result in higher embryo recovery on day 4 of pregnancy (current study) or in higher litter size on day of delivery (previous study [19]). This suggests that some fertilized ova were lost during the preimplantation period in obese dams. On the other hand, survivor embryos developing in the reproductive tract of obese females showed significantly higher cleavage rates. Still, blastocysts derived from such embryos showed an elevated incidence of apoptotic cell death [11], altered gene expression [15], and development of insulin resistance [16].

Limitations of the study: Since reproductive physiology and the overweight scoring system (% of body fat vs. BMI) differ between rodents and human, inter-species extrapolations are limited. Since sperm quality analysis was not performed during selection of male mice, the impact of male subfertility on the results cannot be excluded.

## Figures and Tables

**Figure 1 ijerph-19-07918-f001:**
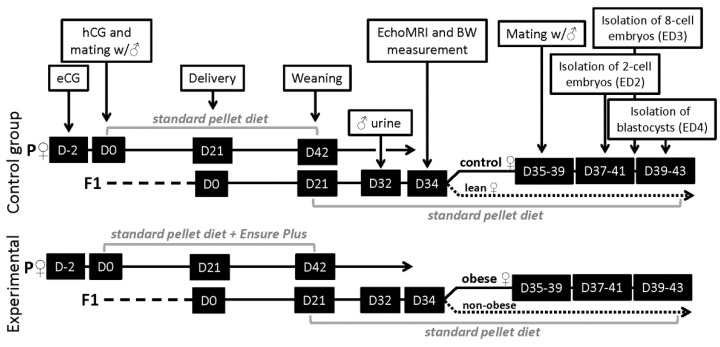
Mouse intergenerational model of diet-induced obesity: Schematic overview of breeding protocol. P, parental generation; F1, filial generation; eCG, pregnant mare’s serum gonadotropin (5 IU ip); hCG, human chorionic gonadotropin (4 IU ip); EchoMRI, non-invasive nuclear magnetic resonance instrument; BW, body weight; ED2, ED3, ED4, embryonic day 2, 3, and 4. Standard pellet diet: 225 g/kg of crude protein, 27 g/kg of crude fat, 30 g/kg of crude fiber, 61 g/kg of ash, and 58 g/kg of saccharides (M1, 3.2 kcal/g). Ensure Plus: high-energy nutritional product (Abbot Laboratories, 1.5 kcal/mL).

**Figure 2 ijerph-19-07918-f002:**
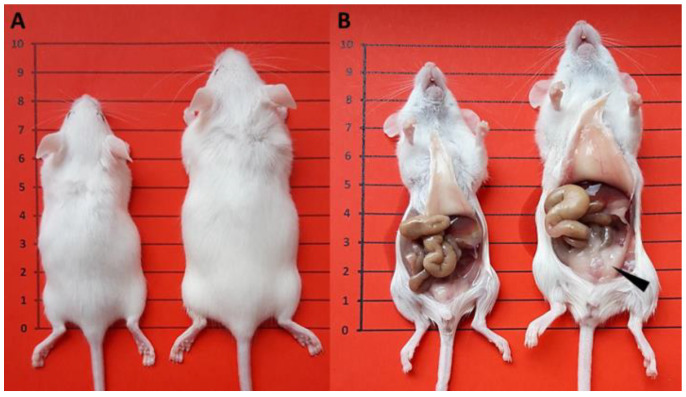
Photographs of control (**left**) and obese mouse females (**right**) developed in an intergenerational obesity model at the age of 39 days. (**A**) Mice sacrificed by cervical dislocation; (**B**) post-mortally dissected mice. Control mice (*n* = 319), derived from dams fed a standard diet, showed physiological body weight (20.47 ± 0.13) and body fat (7.48 ± 0.03). Obese mice (*n* = 327), derived from dams fed a high-energy diet, showed significantly elevated body weight (24.38 ± 0.13) and body fat (12.33 ± 0.09), with massive fat deposits in the abdominal and peri-renal areas (arrowhead). Statistical analysis: *p* < 0.001 for all cases; Student’s *t* test for body weight, Mann–Whitney test for body fat.

**Table 1 ijerph-19-07918-t001:** Effect of female overweight status on ovulation and fertilization rate.

Day	Variable	Control Mice	Obese Mice	OR	95% CI	*p* Value
		Number (%)	Number (%)			
ED2	Mated females	176 (100%)	201 (100%)	n.a.	n.a.	n.a.
	Ovulated females	156 (88.64%)	180 (89.55%	1.10	0.54 to 2.22	0.8686
	Fertilized females	144 (92.31%)	157 (87.22%	0.57	0.25 to 1.24	0.1530
ED3	Mated females	50 (100%)	44 (100%)	n.a.	n.a.	n.a.
	Ovulated females	42 (84.00%)	39 (88.64%)	1.49	0.39 to 6.26	0.5638
	Fertilized females	36 (85.71%)	36 (92.31%)	2.00	0.39 to 13.20	0.4848
ED4	Mated females	93 (100%)	82 (100%)	n.a.	n.a.	n.a.
	Ovulated females	89 (95.70%)	78 (95.12%)	0.88	0.16 to 4.88	>0.9999
	Fertilized females	79 (88.76%)	68 (87.18%)	0.86	0.30 to 2.46	0.8139

Data were analyzed by Fisher’s exact test; Abbreviations: ED, embryonic day; OR, odds ratio; 95% CI, confidence interval; *p*, probability; n.a., not available.

**Table 2 ijerph-19-07918-t002:** Univariate Poisson regression for the number of recovered ova and embryos on embryonic day 2, 3, and 4 in fertilized female mice (with at least one embryo in the reproductive tract).

Day	Variable	Median (IQR)	B	SE	95% CI	*p* Value
ED2	(Intercept)	10.5 (4)	2.316	0.026	9.63 to 10.67	n.a.
	Overweight	12 (3)	0.118	0.035	1.05 to 1.21	0.0008
ED3	(Intercept)	10 (2.5)	2.321	0.052	9.20 to 11.29	n.a.
	Overweight	11 (3)	0.108	0.071	0.97 to 1.28	0.1315
ED4	(Intercept)	11 (3)	2.344	0.034	9.74 to 11.17	n.a.
	Overweight	11 (3)	0.069	0.050	0.97 to 1.18	0.1698

Abbreviations: ED, embryonic day; IQR, interquartile range; B, regression coefficient; SE, Standard Error; 95% CI, 95% confidence interval; *p*, probability; n.a., not available.

**Table 3 ijerph-19-07918-t003:** Univariate Poisson regression for the number of recovered ≥2 cell embryos and for the number of recovered ova (oocytes & degenerates) on embryonic day 2 in fertilized female mice (i.e., mated and ovulating females with at least one embryo in the reproductive tract).

Number of Recovered	Variable	Median (IQR)	B	SE	95% CI	*p* Value
≥2 cell embryos	(Intercept)	10 (4.5)	2.263	0.026	9.12 to 10.14	n.a.
	Overweight	11 (3)	0.109	0.036	1.04 to 1.20	0.0026
Ova	(Intercept)	0 (0)	−0.652	0.115	0.42 to 0.65	n.a.
	Overweight	0 (0)	0.278	0.150	0.98 to 1.77	0.0624

Abbreviations: IQR, interquartile range; B, regression coefficient; SE, Standard Error; 95% CI, 95% confidence interval; *p*, probability; n.a., not available.

**Table 4 ijerph-19-07918-t004:** Univariate Poisson regression for the number of recovered ≥8 cell embryos and for the number of recovered <8 cell embryos & ova on embryonic day 3 in fertilized female mice (i.e., mated and ovulating females with at least one embryo in the reproductive tract).

Number of Recovered	Variable	Median (IQR)	B	SE	95% CI	*p* Value
≥8 cell embryos	(Intercept)	4 (4)	1.433	0.081	3.58 to 4.92	n.a.
	Overweight	5.5 (5.5)	0.357	0.106	1.16 to 1.76	0.0007
<8 cell embryos & ova	(Intercept)	6 (3.5)	1.791	0.068	5.25 to 6.86	n.a.
	Overweight	5.5 (6.5)	−0.112	0.099	0.74 to 1.08	0.2553

Abbreviations: IQR, interquartile range; B, regression coefficient; SE, Standard Error; 95% CI, 95% confidence interval; *p*, probability.

**Table 5 ijerph-19-07918-t005:** Univariate Poisson regression for the number of recovered blastocysts and for the number of recovered morulas & ≤16 cell embryos & degenerates on embryonic day 4 in fertilized female mice (i.e., mated and ovulating females with at least one embryo in the reproductive tract).

Number of Recovered	Variable	Median (IQR)	B	SE	95% CI	*p* Value
Blastocysts	(Intercept)	8 (5)	2.096	0.039	7.53 to 8.79	n.a.
	Overweight	10 (3)	0.166	0.055	1.06 to 1.32	0.0027
Morulas & ≤16 cells	(Intercept)	2 (3)	0.829	0.074	1.98 to 2.65	n.a.
	Overweight	1 (2)	−0.385	0.122	0.54 to 0.86	0.0014

Abbreviations: IQR, interquartile range; B, regression coefficient; SE, Standard Error; 95% CI, 95% confidence interval; *p*, probability; n.a., not available.

## Data Availability

Not applicable.

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
