# Peer review of "Overweight and Fertility: What We Can Learn from an Intergenerational Mouse Obesity Model"

_ijerph, 2022, doi:10.3390/ijerph19137918_

Round 1

Reviewer 1 Report

Review of “Overweight and fertility: What we can learn from an intergenerational mouse obesity model” by Dušan Fabian et al 2022

 The authors of “Overweight and fertility: What we can learn from an intergenerational mouse obesity model” described an extensive study regarding developed in an intergenerational model of obesity induction which eliminates the impact of aging and high-fat feeding. The study design and procedures seem appropriate but would increase the quality of the study if authors would present in parallel the average quality mice sperm.

 The results reported showed the advantages of this approach and discussion was well-founded. To complement the discussion, I just suggest discussing the limitations of this study.

Author Response

The authors of “Overweight and fertility: What we can learn from an intergenerational mouse obesity model” described an extensive study regarding developed in an intergenerational model of obesity induction which eliminates the impact of aging and high-fat feeding. The study design and procedures seem appropriate but would increase the quality of the study if authors would present in parallel the average quality mice sperm.

Reply: We thank for the comment. We agree with the reviewer, that analysis of sperm quality would lower impact of the male subfertility on the results and improve the quality of the study. However, as written in the manuscript, provided data originate from experiments performed between years 2011 and 2020, thus the majority of used mouse males are already excluded from breeding or dead. There is a possibility to provide sperm quality analysis on a cohort of new mouse males of the same strain, age and weight ordered from original provider (Velaz, Czech Republic); however, the relevancy of such analysis for presented results would be doubtful. Furthermore, as mentioned in the Materials and Methods section, males were already selected and only those with previously proven fertility were used for mating purposes. (Fertility was evidenced by ability to provide successful mating leading to embryo production or offspring delivery.) Naturally, such approach is insufficient for disclosure of male subfertility. Thus, in response to reviewer’s comment, drawback was listed between the limitations of the study.

The results reported showed the advantages of this approach and discussion was well-founded. To complement the discussion, I just suggest discussing the limitations of this study.

Reply: In response to reviewer’s comment, a following list of limitations was provided at the end of Conclusions section:

“Since reproductive physiology and the overweight scoring system (% of body fat vs. BMI) differ between rodents and human, inter-species extrapolations are limited. Since sperm quality analysis was not performed during selection of male mice, the impact of male subfertility on the results cannot be excluded. ” (Page 10) 

Reviewer 2 Report

The manuscript entitled “Overweight and fertility: What we can learn from an intergenerational mouse obesity model” by Fabian et al. has some solid data from a very large series of mouse experiments using a novel model of obesity, one in which the mice are fed the same diet as controls but are obese due to (presumably) epigenetic differences during gestation. This expands on the Authors’ previous work on intergenerational obesity models from 2014. The Authors found that obesity increased the number of >8-cell embryos on ED3 and blastocysts on ED4. The Authors do discuss their findings and place them in context: some studies show obesity decreasing oocyte and embryo number, others show no difference; and the increased number of >8 cell embryos and blastocysts doesn’t imply that the embryos or blastocysts are healthy or of high quality. I have some general critiques and concerns listed below. Much remains to be done in these experiments, but they are out of the scope of a small paper focused on the effect of obesity on oocyte and embryo number.

·      English needs work

·      Please divide mouse methods section with more headers describing the particular assays/experiments. i.e. lines 127-135 should be called something like “Classification of mice into Mated, Ovulating, and Fertilized” and contain very detailed description of what each classification is.

·      Figure 2 legend doesn’t reference A or B in the figure

·      Suggestion: in each table/figure legend, define the categories of mice (mated, ovulated have either/both oocytes and embryos, fertilized have at least one embryo) as well as the abbreviations.

·      Since the Authors make a distinction between unfertilized oocytes and degenerate oocytes, could they present this data in the paper?

·      Why the differences in number between E2, E3, and E4 mice? E2 is >300, E3 is <100, and E4 is >150.

Author Response

The manuscript entitled “Overweight and fertility: What we can learn from an intergenerational mouse obesity model” by Fabian et al. has some solid data from a very large series of mouse experiments using a novel model of obesity, one in which the mice are fed the same diet as controls but are obese due to (presumably) epigenetic differences during gestation. This expands on the Authors’ previous work on intergenerational obesity models from 2014. The Authors found that obesity increased the number of >8-cell embryos on ED3 and blastocysts on ED4. The Authors do discuss their findings and place them in context: some studies show obesity decreasing oocyte and embryo number, others show no difference; and the increased number of >8 cell embryos and blastocysts doesn’t imply that the embryos or blastocysts are healthy or of high quality. I have some general critiques and concerns listed below. Much remains to be done in these experiments, but they are out of the scope of a small paper focused on the effect of obesity on oocyte and embryo number.

Reply: We thank reviewer for the valuable comments and understanding. We agree that there are numbers of unsolved questions left and work to be done. However, without the presentation of the impact of maternal overweight on qualitative reproductive parameters, the general picture wouldn’t be complex.

  • English needs work

Reply: As written in the manuscript, the first version of the text was submitted to Andrew Billingham, a native-speaking English teacher, who provided revision of English spelling, grammar and syntax. To improve the quality of language, revised manuscript was submitted to Language Editing Services of MDPI. The majority of suggested changes was accepted, except the change in the title. Please, see enclosed certificate.

  • Please divide mouse methods section with more headers describing the particular assays / experiments. i.e. lines 127-135 should be called something like “Classification of mice into Mated, Ovulating, and Fertilized” and contain very detailed description of what each classification is.

Reply: Materials and Methods section was divided into several sub-sections with sub-headings according to reviewer’s suggestion (Please, see pages 2-4). Classification of mouse females was described in more detail as follows:

“Female mice with a vaginal sperm plug were classified as “mated females”. Female mice with developing corpora lutea and at least one ova or embryo in the reproductive tract were classified as “ovulated females”. Female mice with at least one ≥2 cell embryo in the reproductive tract were classified as “fertilized females”.” (Page 4)

  • Figure 2 legend doesn’t reference A or B in the figure

Reply: Following reference to panels A and B was added to the Figure 2 legend: “(a) Mice sacrificed by cervical dislocation; (b) Post-mortally dissected mice.” (Page 5)

  • Suggestion: in each table/figure legend, define the categories of mice (mated, ovulated have either/both oocytes and embryos, fertilized have at least one embryo) as well as the abbreviations.

Reply: Categories of mice and un-explained abbreviations were defined in the table legends. Please, see tables 1 to 3 (Pages 5-7). 

  • Since the Authors make a distinction between unfertilized oocytes and degenerate oocytes, could they present this data in the paper?

Reply: During statistical analysis of collected data, two approaches were performed: 1. analysis of raw data with separate groups of unfertilized oocytes, degenerated oocytes, embryos at particular stages, etc., and 2. analysis of data with joined groups of oocytes, cleaving embryos, etc. Since both approaches resulted in almost identical outcomes, we decided to present results of second approach with joined groups, as it seemed more understandable for the reader. Another reason for presentation of joined groups was that we were not able to exactly differentiate degenerated oocytes from degenerated embryos on ED4. Anyway, we agree with the reviewer that available data on unfertilized oocytes and degenerate oocytes should be shown in the manuscript. Thus, following paragraph was added to the Results section:

“In fertilized female mice, the majority of oocytes recovered on ED2 and ED3 showed normal morphology. The occurrence of degenerated oocytes was very rare (control group: 0.013% on ED2, 0.008% on ED3; overweight group: 0.008% on ED2, 0.005% on ED3). On ED4, the proportion of degenerated oocytes/embryos reached 0.047% in the control group and 0.042% in the overweight group. No statistical difference between the groups was detected on any embryonic day.” (Page 6)   

  • Why the differences in number between E2, E3, and E4 mice? E2 is >300, E3 is <100, and E4 is >150.

 Reply: As mentioned in the manuscript, data for statistical analysis were collected during 9 years. Between 2011 and 2020, production of overweight mouse females (based on specific breeding protocol shown in Fig. 1) was repeated 14 times. After natural mating, the highest number of overweight females (together with equivalent number of controls) was subjected to embryo isolation on day 2 of pregnancy (collected 2-cell embryos were used in several series of in vitro experiments). A lower number of overweight/control females was subjected to embryo isolation on day 4 of pregnancy (collected blastocysts were used in series of experiments focused on their qualitative analysis). Finally, the lowest number of overweight/control females was subjected to embryo isolation on day 3 of pregnancy (collected cleaving embryos were used in series of experiments focused on developmental dynamics). The numbers of animals in each experiment were adjusted in accordance with the principles of the 3Rs. Limit was determined by minimal amount of embryonic material necessary for adequate statistical analysis. Thus, specific aims of particular experiments lead to disproportion in final numbers of embryo donors presented in submitted overviewing study. In response to reviewer’s comment, following paragraph was added to the Materials and Methods section:

“Data gathering was performed between the years 2011 and 2020. During this period, ova/embryo isolation was performed eight times on ED2, three times on ED3, and five times on ED4.” (Page 3/4)

Reviewer 3 Report

I recommend that this article be published because I found the evaluation of the effects of obesity on the ability to conceive, fertilization rate, and in vivo development of embryos in regularly cycling, spontaneously ovulating, and naturally mated mouse females based on statistical analysis of the data collected interesting.

Author Response

I recommend that this article be published because I found the evaluation of the effects of obesity on the ability to conceive, fertilization rate, and in vivo development of embryos in regularly cycling, spontaneously ovulating, and naturally mated mouse females based on statistical analysis of the data collected interesting.

Reply: We thank reviewer of for positive evaluation of the manuscript.

Round 2

Reviewer 2 Report

The Authors have addressed all of my concerns.